Analysis of the light intensity dependence of the growth of Synechocystis and of the light distribution in a photobioreactor energized by 635 nm light

http://orcid.org/0000-0002-6325-6842 Cordara Alessandro 1 2
Re Angela 2 angela.re@iit.it
Pagliano Cristina 1
http://orcid.org/0000-0002-1311-8118 Van Alphen Pascal 3
Pirone Raffaele 4
Saracco Guido 2 5
http://orcid.org/0000-0002-4268-8080 Branco dos Santos Filipe 3
http://orcid.org/0000-0002-6652-4937 Hellingwerf Klaas 3
Vasile Nicolò 2
1 Applied Science and Technology Department—Biosolar Lab, Politecnico di Torino , Turin , Italy
2 Centre for Sustainable Future Technologies, Istituto Italiano di Tecnologia , Turin , Italy
3 Swammerdam Institute for Life Sciences, University of Amsterdam , Amsterdam , Netherlands
4 Applied Science and Technology Department, Politecnico di Torino , Turin , Italy
Eroglu Ela
5 Current affiliation: Applied Science and Technology Department, Politecnico di Torino, Turin, Italy

Electronic publication date: 2018 Jul 27
Publication date: 2018
Volume: 6
Electronic Location ID: e5256
Received 2018 Apr 25; Accepted 2018 Jun 26
Copyright: © 2018 Cordara et al.
Copyright year: 2018
Copyright holder: Cordara et al.
License: This is an open access article distributed under the terms of the Creative Commons Attribution License, which permits unrestricted use, distribution, reproduction and adaptation in any medium and for any purpose provided that it is properly attributed. For attribution, the original author(s), title, publication source (PeerJ) and either DOI or URL of the article must be cited.
License URL: https://creativecommons.org/licenses/by/4.0/

Keywords: Modelling, Photoinhibition, Cyanobacteria, Photobioreactor, Lightening conditions

Funding: European Union’s Horizon 2020 research and innovation programme 760994 The research leading to this publication has received funding from the European Union’s Horizon 2020 research and innovation programme under grant agreement No 760994 (ENGICOIN project). The funders had no role in study design, data collection and analysis, decision to publish, or preparation of the manuscript.

==============================
Synechocystis gathered momentum in modelling studies and biotechnological applications owing to multiple factors like fast growth, ability to fix carbon dioxide into valuable products, and the relative ease of genetic manipulation. Synechocystis physiology and metabolism, and consequently, the productivity of Synechocystis-based photobioreactors (PBRs), are heavily light modulated. Here, we set up a turbidostat-controlled lab-scale cultivation system in order to study the influence of varying orange–red light intensities on Synechocystis growth characteristics and photosynthetic activity. Synechocystis growth and photosynthetic activity were found to raise as supplied light intensity increased up to 500 μmol photons m−2 s−1 and to enter the photoinhibition state only at 800 μmol photons m−2 s−1. Interestingly, reverting the light to a non-photo-inhibiting intensity unveiled Synechocystis to be able to promptly recover. Furthermore, our characterization displayed a clear correlation between variations in growth rate and cell size, extending a phenomenon previously observed in other cyanobacteria. Further, we applied a modelling approach to simulate the effects produced by varying the incident light intensity on its local distribution within the PBR vessel. Our model simulations suggested that the photosynthetic activity of Synechocystis could be enhanced by finely regulating the intensity of the light incident on the PBR in order to prevent cells from experiencing light-induced stress and induce their exploitation of areas of different local light intensity formed in the vessel. In the latter case, the heterogeneous distribution of the local light intensity would allow Synechocystis for an optimized usage of light.

Introduction

Synechocystis sp. PCC6803 (hereafter Synechocystis) was the first photosynthetic organism to have its genome fully sequenced (Kaneko et al., 1996). A wealth of transcriptomic (Anfelt et al., 2013; Beck et al., 2014; Angermayr et al., 2016), proteomic (Fang et al., 2016) and metabolomic (Yang, Hua & Shimizu, 2002; Yoshikawa et al., 2013) studies allowed to investigate Synechocystis regulatory, signalling and metabolic pathways in finer details than in any other cyanobacterium. Synechocystis has attracted much interest as model organism in product-oriented industrial biotechnology due to the ability to effortlessly recycle carbon dioxide (CO2) into valuable fuels and chemicals, the simplicity of its culture conditions, the ease of genetic manipulation and its relatively fast cell growth compared to higher plants (Janssen et al., 2003; Angermayr, Hellingwerf & Teixeira De Mattos, 2009). Genetic engineering of cyanobacteria has demonstrated the opportunity to channel solar energy into the formation of various commodity products (Angermayr, Gorchs Rovira & Hellingwerf, 2015; Zhou et al., 2016). In the last decades, Synechocystis has served in many genetic engineering studies as biofactory for the production of a variety of products (Yu et al., 2013; Singh et al., 2017), such as ethanol (Gao et al., 2012), isobutanol (Varman et al., 2013), lactate (Angermayr, Paszota & Hellingwerf, 2012; Joseph et al., 2013) and polyhydroxyalkanoate (Luengo et al., 2003), which can be widely utilized in biotechnology and industrial fields.

The intensive exploitation of this microorganism for industrial uses strongly depends on the choice of optimal growth conditions, main operational parameters including culture density (Esteves-Ferreira et al., 2017; Straka & Rittmann, 2018), pH (Touloupakis et al., 2016), temperature (Panda et al., 2006), mixing rate and light environment (Touloupakis et al., 2016; Singh et al., 2009). Even though extensive investigation showed that Synechocystis productivity is sensitive to most of the aforementioned operational parameters (Yu et al., 2013; Burrows et al., 2009; Nanjo et al., 2010; Chaves, Kirst & Melis, 2015), it is undoubted that productivity is tightly coupled with the light absorption efficiency of optical energy conversion systems. Therefore, light management during Synechocystis cultivation in photobioreactors (PBRs) is by far the most remarkable factor to account for in order to boost the practical exploitation of this microorganism. Synechocystis is able to absorb energy across the visible spectrum, mainly through three classes of pigments: bilins (Gan & Bryant, 2015), chlorophyll a (Chl a), which is associated with Photosystem II (PSII) and Photosystem I reaction centre cores (Vermaas, 1996), and carotenoids (Glazer, 1977; Colyer et al., 2005). Achieving high performance in PBRs requires high intensity light, which nonetheless can cause light associated damages. A number of studies sought to dilute the supplied light intensity by optimizing the light spectra distribution. These attempts were encouraged by the integration of light emitting diodes (LEDs) within indoor PBRs (Ooms et al., 2017). Tailoring light wavelength spectrum affords improved growth conditions stability and reproducibility and has been shown to lead to concrete achievements in biomass productivity and ultimately in the accumulation of useful products. For example, a perfect fit of the red light with the absorption peak of the Chl a and phycocyanobilin was observed to lead to an increased growth in cyanobacteria during cultivation (Wyman & Fay, 1986; Wang, Fu & Liu, 2007; Alphen, 2018). In other cases, dynamic adjustments of light wavelength during the cultivation of Chlorella vulgaris and Haematococcus pluvialis allowed the increase of microorganism productivity (Katsuda et al., 2004).

It is well known that the exposure of photosynthetic organisms to strong solar irradiation results in inhibition of the electron transfer activity of PSII, referred to as photoinhibition responses (Powles, 1984). This phenomenon derives from an imbalance between the photodamage brought to PSII and the repair mechanisms for such damage (Murata et al., 2007). Despite the numerous studies conducted on this topic highlight that the main target of photoinhibition is the D1 protein of PSII reaction centre (Tyystjärvi, 2013), the molecular mechanisms of PSII photoinhibition are not yet completely understood. Upon exposure of photosynthetic organisms to strong irradiation, two mechanisms contribute to the photodamage of PSII, which are called acceptor-side and donor-side photoinhibition. In the acceptor-side mechanism, strong illumination causes the over-reduction of PSII, due to the double reduction of the primary quinone acceptor (QA) that, in such condition, can no longer serve as an electron carrier. The recombination between the doubled reduced form of QA and the primary radical pair P680+ and Pheo− leads to the formation of the triplet state of the P680, which can react with molecular oxygen leading to generation of the reactive form of oxygen (ROS) singlet oxygen (1O2) (Mulo et al., 1998; Vass et al., 1992). Due to the extremely short lifetime of this ROS, 1O2 is thought to impair mainly the proteins and lipids nearby its production site (Triantaphylidès & Havaux, 2009). Conversely, the donor-side photoinhibition is not mediated by ROS and occurs when the reduction of the PSII is slower than its oxidation, due to inactivation of the oxygen evolving system. This leads to an extended lifetime of the radicals TyrZ+ and P680+ that act as strong oxidants against the surrounding proteins and lipids, resulting in a damage to the PSII (Bumann & Oesterhelt, 1995).

In this work, we investigated the effect of increasing intensities (in the range 50–1,460 μmol photons m−2 s−1) of orange–red light on the autotrophic growth of Synechocystis in a turbidostat-controlled lab scale PBR. Monitoring Synechocystis physiological state under varying light regimes, we found that growth rate, cell size and PSII activity were influenced by light intensity, albeit in slightly different ways. Synechocystis cells proved to be resilient to high light stress conditions, suffering photoinhibition only above 800 μmol photons m−2 s−1, and showed a remarkable ability to recover from the complete state of photoinhibition experienced at 1,460 μmol photons m−2 s−1 when reverting light to 200 μmol photons m−2 s−1. Further, we combined the experimental analyses with system modelling and the related multi-physics analysis to investigate the influence of local light intensity distribution on photoinhibition of microorganism.

Materials and Methods

Strain and preculture conditions

For all experiments, we used wild-type Synechocystis sp. PCC 6803, a glucose-tolerant derivative kindly provided by Devaki Bhaya (Department of Plant Biology, Carnegie Institution for Science, Stanford, CA, USA). The cells were grown in flasks in 25 ml of BG11 medium (Stanier et al., 1971) with a modified protocol as described in Van Alphen & Hellingwerf (2015). Precultures were grown for 4 days at 30 °C in a shaking incubator at 120 rpm (Innova 44, New Brunswick Scientific, Edison, NJ, United States) under constant illumination of orange–red (632 nm) and blue (451 nm) light (10:1 photon ratio) at 30 μmol photons m−2 s−1, measured with a LI-250 quantum sensor (LI-COR, Lincoln, NE USA).

PBR growth conditions

A Synechocystis preculture was used to seed the PBR. The culture was grown in a flat panel PBR model FMT150.2/400 Photon System Instruments (Nedbal et al., 2008) in a final volume of approximately 380 ml in the BG-11 medium modified as described above, supplemented with 10 mM of NaHCO3.

The PBR (Fig. 1) is provided with a combined pH/temperature probe, a Clark-type dissolved O2 (dO2) probe (all probes Mattler-Toledo), and an integrated densitometer that measures the optical density (OD) at 720 and 680 nm.

Figure 1 Photobioreactor schematic representation.

(A) Body of the flat panel PBR FMT150.2/400 composed of a 390 ml transparent removable flat vessel. On top of the vessel, a stainless lid accommodates different tubes, connectors and sensors. The base of the instrument contains a control unit with analogic and digital electronic circuits. Enlarged the details of the red and blue LEDs installed in the light panel of the reactor, the densitometer and the fluorometer. (B) Red and blue LED spectra of PBR FMT150.2/400. (C) Transmission spectrum of cyanobacterial culture affected by light absorptions, light scattering. The lines and arrows indicate wavelength of the light sources present in the flat panel reactor and the detection range of the detector filter. (D) 3D modelled geometry of PBR with modelled domains selection: 1-closing, 2-inoculum, 3-sparger, 4-air, 5-sampling, 6-culture, 7-stirring bar, 8-wall of vessel, 9-base of vessel.

The cyanobacterial suspension was illuminated from one side with orange–red light (636 nm) by high-power LEDs. The light regimes applied by the LED board provided cells with the following light intensities as measured outside the PBR, opposite of and at the centre of the light panel: 50, 200, 300, 500, 800, 950 and 1,460 μmol photons m−2 s−1. The cells were subjected to increasing light intensity every 24 h. The 24 h period of light acclimation was sufficient for establishing stable (variance <1%) growth rate and dissolved oxygen in the culture medium for each light intensity except for the light regime of 1,460 μmol photons m−2 s−1, where stable values were not obtained. The temperature and pH were kept constant at 30 °C and 8.0, respectively, by automatically adjusting pCO2 using a gas mixing system GMS150 (Photon Systems Instruments, Drásov, Czech Republic). CO2 was provided in a mixture with N2 with a gas flow of 150 ml min−1 controlled by a mass flow controller (Smart Mass Flow Model 5850S; Brooks Instruments, Hatfield, PA, United States). The PBR was run in turbidostat mode with the OD720, measured by the integrated densitometer, calibrated to the bench-top spectrophotometer OD730 to maintain the OD730 at approximately 0.4 (turbidostat set to a maximum deviation of 3%) at 50 μmol photons m−2 s−1. The turbidostat mode allowed the culture to hold cell density constant and to remain in exponential phase under all tested light conditions. Dissolved molecular oxygen was normalized to the values obtained at 50 μmol photons m−2 s−1.

Dry weight, cell size and count measurements

At the end of each 24 h step of light increment, 22 ml of culture were harvested to perform in parallel dry weight measurements and cell size analysis.

For the determination of dry cell weight, cellulose acetate membranes (0.2 μm, Whatman, Maidstone, United Kingdom) were washed with milli-Q water (Merck Millipore Reference, Burlington, MA, United States), left to dry for 24 h at 90 °C in a stove (Electrolux, Stockholm, Sweden) and weighted with an analytical balance (AB204; Mettler Toledo, Columbus, OH, United States). Subsequently, the membranes were used for filtering 20 ml of sampled culture. After washing once with milli-Q water to remove salts, the membrane filter was left to dry overnight in the stove at 90 °C and finally weighted again. In parallel, the OD730 of the sampled cells was measured with a spectrophotometer (Lightwave II; Biochrom, Cambridge, United Kingdom) and used to normalize the dry cell weight per OD730.

The average cell size and cell number were measured with the CASY counter instrument (Roche Applied Science, Penzberg, Germany). A volume of 20 μl of harvested culture was diluted with 10 ml of CASY ton solution. The average cell size was measured working in a range of calibration between zero and five μm with a capillary of 60 μm.

Model description

The 3D multi-physics model of the PBR was developed on the COMSOL 5.3® platform and allowed us to simulate different phenomena such as fluid dynamic, light transmission in different media, cyanobacterial growth kinetics and mass transfer by formulating the corresponding equations. Figure 1D shows the design of the 3D model based on the reactor geometry. Free tetrahedral meshing was applied to the created model prior to analysis. Meshing size was selected in order to prevent inaccuracy and imprecision of modelling resulting from model meshing.

Assumptions, inlet and boundary conditions

To solve the different mass balances and kinetic equations, it is necessary either to state the initial and boundary conditions, which include inlet, outlet and wall conditions, or to discuss the different assumptions, which clarify the limitations of the created model: the inlet velocity of recycling gas was measured experimentally. As there is no liquid exchanging during experiments, the liquid velocity in the inlet/outlet area is equal to 0;

the mass transfer between liquid culture and gas is considered;

all conditions in the model of the PBR, where both liquid and gas phases are present, were formulated by assuming the gas flux at the reactor boundaries negligible and by setting the liquid velocity at the reactor surface wall different from zero. Slip conditions were applied to all PBR walls;

since only CO2/N2 gas mixture flows through the sparger, its fluid-dynamic model assumed a single phase to be present. Therefore, the wall conditions used for gas flow were slip conditions, which assume that the gas velocity at the sparger solid surface is calculated by the mathematical model;

the working conditions have to be furnished to the model: temperature, inlet gas velocity (or flow rate), initial, inlet and outlet pressure, amount of initial microorganisms and nutrients;

the influence of nutrient concentration, temperature and light intensity on cyanobacterial growth rate are all considered.

Mathematical model

The mathematical model is described in the following sections. All the variables and parameters employed in the equations are listed and described in the nomenclature as well as in Table S1. To estimate the necessary parameters, we relied on experimentally determined values or on an inference procedure by fitting model simulations to observed data. Specifically, we compared simulated and experimental data using a statistical analysis of errors based on the Levenberg–Marquardt method, coupled with the second least-squares analysis. Since the method implements a constrained search procedure, it requires specifying lower and upper bounds on the unknown parameters, which were selected within the ranges of values most frequently observed in literature. The parameter space exploration stopped when the model simulation best fit the experimental data.

Fluid-dynamic equations

We generally need to model different domains inside the bioreactor: a gas-liquid mixture inside the vessel, and a single gas phase inside the sparger domain. Therefore, the continuity equations and the momentum balance equations need to be adapted depending on the modelled domain.

The general double-phase fluid dynamic continuity equations are formulated through Eqs. ((1)–(3)) and allow to account for the coexistence of the bubble gas phase (dispersed phase, named by ‘d’ as subscript) and the liquid phase (continuous phase, named by ‘c’ as subscript): (1) (ρc−ρd)[∇⋅(Φd(1−cd)uslip−Dmd∇⋅Φd)+mdcρd]+ρc(∇⋅u)=0

(2) u=ϕcρcuc+ϕdρdudρ

(3) ∅c+∅d=1

where Eq. (1) is the continuity equation, Eq. (2) expresses the velocity vector u, Φc and Φd are the volume fractions corresponding to liquid and gas phases respectively, and ρ is the pseudo-continuous phase density. Equation (3) describes the relation between the volume fractions for the continuous Φc and dispersed Φd phases.

As to the sparger, since a single gas phase exists, we modelled the fluid dynamic variables trends accounting only for the terms of the (Eq. (1)) related to the gas phase, which is expressed by the following equation: (4) ∂ρg∂t+∇⋅(ρu)g=0

Regarding the momentum balance equations, we adopted the Navier–Stokes model for the liquid–gas multiphase system, by using the following formulation (Li, Hu & Liu, 2014) (5) ∂(ρu)∂t+∇⋅(ρuu)=−∇p−∇⋅τ+ρg+F

(6) τ=−μ[(∇u+(∇u)T)−23(∇⋅u)I]

where p is the pressure, g is the gravity acceleration vector, τ is the stress tensor and μ is the effective viscosity. For the single-phase sparger model, the momentum balance equations are similar to Eq. (5), where we replaced the effective viscosity with the gas viscosity.

In Eq. (5), the effective viscosity includes not only the molecular viscosity μi but also turbulent viscosity μi,T (Luo & Al-Dahhan, 2011), which accounts for the influence exerted by the turbulent flow Eq. (7).

Among the several models introduced to handle the turbulent viscosity, we adopted the standard k–ε model to simulate the turbulent flow of the fluid entrained with cyanobacteria in the mechanically stirred PBR. The standard k–ε turbulent model is computationally stable, even in the presence of complex physics, and is applicable to a wide variety of turbulent flows.

The equations to calculate the effective viscosity within the k–ε model are listed below: (7) μeff=μi+μi,T

(8) μT=ρCμk2ε

where Cμ is a model constant equals to 0.09 and k is the turbulence kinetic energy which can be calculated by the transport equations (Ali, 2014)(9) ρ∂k∂t+ρu⋅∇k=∇⋅((μ+μTσk)∇k)+Pk−ρε

where σk is a model constant equals to 1.0, derived from (Wilcox, 1993). To obtain the Pk values we used Eq. (10) (10) Pk=μT(∇u(∇u+(∇u)T)−23(∇⋅u)2)−23ρk∇⋅u

where the value of turbulent energy dissipation rate ε was calculated by Eq. (11): (11) ρ∂ε∂t+ρu⋅∇ε=∇⋅((μ+μTσε)∇ε)+Cε1εkPk−Cε2ρε2k

In this equation, Cε1 and Cε2 are constants equal to 1.44 and 1.92, respectively. Finally, in a turbulent bubbly flow model, the difference between gas velocity and liquid velocity consists of two terms: slip velocity and drift velocity (Eq. (12)): (12) ud−uc=uslip−Dmd(1−cd)Φd

The slip velocity uslip represents the relative velocity of the phases and the drift velocity is the additional velocity appearing when turbulence is taken into account. The drift velocity can be calculated by Eq. (13), whereas the slip velocity can be calculated by using a pressure-drag balance (Eq. (14)): (13) udrift=−Dmd(1−cd)∇∅d

(14) 3Cdrag4db|uslip|uslipρl=−∇p

where the drag coefficient Cdrag can be computed by different formulas (Hartmann et al., 2013). In our case, Cd was computed by Eq. (15): (15) Cd=0.622ξgρldb2+0.235

where ζ is the surface tension coefficient and db is the average bubble diameter which was measured as three mm through video imaging (db = 3 mm).

Heat transfer with radiation: light transmission equations

The balance of the radiative intensity, including contributions regarding propagation, emission, absorption and scattering is formulated through the general radiative transfer equation (Modest, 2003) and can be written as follows: (16) Ω⋅∇I(Ω)=κIb(T)−βI(Ω)+σS4π∫4πI(Ω′)Φ(Ω′,Ω)dΩ′

To account for the effect mediated by the bubble volume fraction and cyanobacterial cell concentration, we adapted the Lambert–Beer’s law: (17) II0=exp(−βz)

where I is the local light intensity and I0 represents the incident light intensity, β is the extinction coefficient and z represents the path-length of the light through the material. Equation (17) does not account either for the light scattering by bubbles, which change the direction of light transmission, either for the light absorption by cyanobacteria [51]. Therefore, we modified Eq. (17) as follows: (18) II0=exp(−(βS4+Ka)z)

Equation (18) accounts for light absorption by Synechocystis cells by means of kα, which is the absorption coefficient associated with cyanobacteria, and for the scattering associated with the bubble volume fraction through S, the interfacial area per unit volume which is a function of bubble size and bubble number density. By replacing the interfacial area with the bubble volume fraction ∅d and the bubble diameter, db, Eq. (18) is converted in the equation Eq. (19): (19) II0=exp(−3∅dzdb−Kaz)

The difference between Eqs. (17) and (19) is the effect of cyanobacteria light absorption on local light intensity distribution. The average diameter of bubbles was calculated by using the video imaging technique and performing the procedure described in Ali (2014) and Zhang, Dechatiwongse & Hellgardt (2015). Equation (19) was then used in the general radiative transfer equation.

Calculating the radiative heat source requires information on the temperature regime throughout the entire vessel domain, which is obtained solving the general heat transfer balance equation. The general heat transfer balance (Bird, Stewart & Lightfoot, 2002), which takes into account the radiation in participating media, is expressed in Eq. (20) (20) ρCp∂T∂t+ρCpu⋅∇T+∇⋅q=Q+Qr

where Qr is the radiative heat source expressed as Eq. (21) (21) Qr=κ(G−4πIb)

Kinetic models and calculation theory: cyanobacterial growth equations

Generally, cyanobacterial growth rate is strongly influenced by various factors such as temperature, light intensity and nutrient concentration. Temperature usually affects the activity of enzymes involved in the cellular duplication, whereas light intensity determines the energy that cells can absorb for their maintenance and growth. Nutrient elements including sulphur, carbon, phosphorus and nitrogen are necessary for cyanobacteria to compose their biomass (Dechatiwongse et al., 2014).

The kinetics of cyanobacterial growth is usually defined by the Monod model, Eq. (22), which only considers the effect of nutrient concentration (Vatcheva et al., 2006), because additional environmental parameters such as temperature and light intensity are always kept constant during experiments (Solimeno et al., 2015). In the Monod model, as shown in Eq. (22), the maximum specific growth rate μmax is treated as a constant, but in reality it is a function of light intensity and temperature. When nutrients are in excess, the growth rate is independent of nutrient concentration and expressed as: (22) μgr=μmaxCks+C

where C is the concentration of the limiting substrate for growth. In order to take into account also the effects exerted by nutrients and local light intensity on cyanobacterial growth kinetics, we decided to evaluate a modified Monod equation: the Aiba model (Aiba, 1982).

The Aiba model, shown in Eq. (23), is usually employed to simulate the effect of light intensity on cyanobacterial growth rate: it is capable of modelling the photo-limitation regime under low light intensity, the photo-saturation regime under optimal light intensity, and the photo-inhibition regime under intense light intensity (Zhang et al., 2015). Similarly, the model can also be applied to describe the photo-dependence of the oxygen production rate.

In this equation, (23) μgr=μmax⋅II+ks+I2ki

μmax is the maximum growth rate, and ks and ki refer to the light saturation and photo-inhibition, respectively. These parameters are only dependent on cyanobacterial properties and were fitted from experimental observations.

All the variables and parameters used in the mathematical models are listed and explained in the nomenclature and in Table S1 of Supplemental Information.

Determination of the photosynthetic efficiency

The photosynthetic efficiency was calculated as grams of biomass formed per mol photons. We calculated the amount of light available to the culture as the input to the PBR, which we called Iin, minus the light remaining after the passage through the culture, which we called Iout. Iout was calculated through model simulations to calculate how much light was absorbed in the reactor volume in 1 h. We used the growth rate and dry weight values to calculate how much biomass was produced in these square centimetres times the 2.4 cm depth of the culture for the actual volume in 1 h.

Results

Growth rate, oxygen evolution activity and dimension of cells are dynamically regulated by light intensity

During cultivation, orange–red light was used since it resulted in an optimal light regime for growing Synechocystis in our PBRs system. Conversely, light with wavelengths lower than 580 nm (green–blue) or higher than 670 nm (far-red) proved to be not efficient for its growth (Singh et al., 2009). This knowledge served as a prerequisite to set up experiments aiming to quantitatively evaluate the effects of increasing intensities, ranging between 50 and 1,460 μmol photons m−2 s−1, of red–orange light on the adjustments of the physiological state in Synechocystis. We studied the long-term photoinhibition in Synechocystis grown at increasing light intensities by analysing changes in the growth rate, the physiological parameter of oxygen evolution activity and the cell size at each incremental step of light intensity. By running the PBR in turbidostat mode, Synechocystis was grown in a semi-continuous regime so that cells were constantly maintained in the exponential growth phase.

Firstly, we measured Synechocystis growth rate to evaluate its generation time, and the amount of oxygen dissolved in the medium (dO2), which provides an indication of the PSII activity within cell (Schuurmans et al., 2015). Both the growth rate of Synechocystis (Fig. 2A) and the dissolved oxygen produced by the cells (Fig. 2B) were clearly affected by increasing light intensity. The dynamics of both variables could be broadly partitioned into four phases: an initial phase at 50 μmol photons m−2 s−1 where cells were not photoinhibited, a second phase up to 500 μmol photons m−2 s−1, where Synechocystis cells doubling time and PSII activity attained their maximum values, a third photoinhibitory phase up to 1,460 μmol photons m−2 s−1 where both parameters dropped off, and a final recovery phase at 200 μmol photons m−2 s−1. Hereafter, the photon irradiance of 50 μmol photons m−2 s−1, was considered the control photon irradiance. At this light condition, cells featured a relatively slow metabolism, as evidenced by the modest growth rate of 0.054 ± 0.003 h−1 (corresponding to a doubling time of ≈ 13 h) and a limited level of oxygen dissolved in the medium (roughly 35 μM). This dissolved oxygen concentration at this photon irradiance was used as reference to normalize the measurements at all sampled points (normalized reference value at one). At a photon irradiance of 200 μmol photons m−2 s−1 Synechocystis grew two times faster than in the control light condition, showing a growth rate of 0.114 ± 0.005 h−1 (corresponding to a doubling time of ≈ 6 h), as shown in Fig. 2A. When photon irradiance was increased up to 300 and 500 μmol photons m−2 s−1, the growth rate remained constant yielding values of 0.117 ± 0.006 h−1 and 0.114 ± 0.005 h−1, respectively. Switching photon irradiance from 50 to 200 μmol photons m−2 s−1 led to a higher than two-fold increase in the relative amount of oxygen dissolved in the medium (from 1.00 ± 0.02 to 2.22 ± 0.07, as shown in Fig. 2B), similarly to the trend displayed by growth rate. However, differently from the Synechocystis growth rate, when we increased photon irradiance to 300 μmol photons m−2 s−1, the relative concentration of oxygen kept increasing up to 2.56 ± 0.12. Such difference in growth rate and dO2 within the photon irradiance range from 200 to 300 μmol photons m−2 s−1 suggests that light could exert different effects on the PSII functionality and the doubling time of the microorganism. In particular, the increase in dO2 concentration measured in the cultivation medium could indicate a fine tuning of the light/energy conversion by PSII, whereas the unvaried growth rate observed could result from a limited utilization of the light energy absorbed (Kramer & Evans, 2011). At a photon irradiance of 800 μmol photons m−2 s−1 a substantial decrease was observed for both growth rate (0.1 ± 0.012 h−1 corresponding to a doubling time of ≈ 7 h) and relative amount of dO2 (2.34 ± 0.11), denoting a light intensity where Synechocystis cells started to get photoinhibited. Further increasing the photon irradiance up to 1,460 μmol photons m−2 s−1, Synechocystis growth rate dropped off to 0.043 ± 0.020 h−1 (corresponding to a doubling time of ≈ 16 h), representing roughly half of the maximum growth rate observed in our experimental set up. Furthermore, over the 24 h acclimation period at this extreme light treatment, the growth rate was found to be constantly decreasing thus highlighting a severe state of growth inhibition. Together with the growth rate, the relative dO2 reached its minimum of 1.45 ± 0.23 at 1,460 μmol photons m−2 s−1. These results evidenced severe PSII photodamage caused in Synechocystis by its exposure to high light intensity, in accordance with extensive literature available for cyanobacteria (see review (Murata et al., 2007) and references therein). To test whether, upon photoinhibition, lowering light intensity could recover the growth rate of Synechocystis, a point of recovery was set to the lowest light irradiance at which the maximal growth rate was observed (i.e., 200 μmol photons m−2 s−1). Upon reverting the photon irradiance to 200 μmol photons m−2 s−1, cells showed a remarkable ability to recover completely from the state of photoinhibition, as attested by the increased growth rate up to 0.102 ± 0.009 h−1 (corresponding to a doubling time of ≈ 7 h), which is similar to the maximum growth rate previously measured under the same light condition (Fig. 2A (black dot) and Fig. 3). Similarly, the relative concentration of dO2 in the medium was found to increase up to 2.13 ± 0.24, attesting a full recovery of the PSII activity as well (Fig. 2B (black dot) and Fig. 3). We found that the recovery half-time is approximately 3 h and that both growth rate and dO2 remain stable after recovery during the 24 h.

Figure 2 Challanging Synechocystis by high light intensity revealed its adaptive capacity.

Synechocystis behaviour was assessed by quantifying the growth rate and the relative concentration of dissolved oxygen in the medium under increasing photon irradiance. The turbidostat-controlled cultures were grown at constant temperature (30 °C) and pH (8.0) under orange–red light and acclimated for 24 h at each light intensity. The figure shows Synechocystis ability to fully recover after passing through a complete state of photoinhibition at 1,460 μmol photons m−2 s−1. (A) Growth rate of Synechocystis evaluated at each light intensity. (B) Oxygen released in the medium of the PBR by Synechocystis. The blue dots show the mean values derived from three biological replicates and are accompanied by their respective standard deviation bars. Data were normalized to the values obtained at 50 μmol photons m−2 s−1. The orange triangles show the simulated values according to our PBR model. In both panels (•) stands for point of recovery, which was set at 200 μmol photons m−2 s−1, and the asterisk (*) indicates that no steady state could be reached in 24 h at this condition (as shown by the large error bars).

Figure 3 Recovery of growth rate of Synechocystis after high light treatment.

Synechocystis quickly recovers from high light (1,460 µmol photons m−2 s−1, left side) after reducing light intensity to a non-photoinhibiting intensity (200 µmol photons m−2 s−1, right side). Dissolved oxygen concentration (red line) and growth rate (black squares, calculated per turbidostat cycle) are shown. The half-life of this recovery is 3 h.

As aforementioned, our phenotypic characterization included Synechocystis cell size, which was found to vary across the experimentally tested conditions. Even though the bacterial life cycle is usually the major determinant of morphological traits, including cell size, several studies reported that cyanobacteria modify their morphology to optimize their functionality to exogenous factors in natural contexts (Montgomery, 2015). However, the mechanisms by which light conditions, including light intensity, influence cell morphology, including cell size, are understudied (Pattanaik, Whitaker & Montgomery, 2011). Table 1 shows Synechocystis cell size changes in response to increasing light intensities. At 50 μmol photons m−2 s−1 the cell size was 2.11 ± 0.06 μm. This size is commonly observed in Synechocystis cells cultivated under non-stressful conditions (Du et al., 2016). At a photon irradiance of 500 μmol photons m−2 s−1 Synechocystis cells reached the maximum size of 3.29 ± 0.13 μm while, at the photon irradiance of 800 μmol photons m−2 s−1, which was found to induce photoinhibition, Synechocystis cell size decreased to 3.09 ± 0.23 μm and then gradually to 2.69 ± 0.21 μm at the maximal photoinhibitory photon irradiance of 1,460 μmol photons m−2 s−1. Notably, at the recovery light regime of 200 μmol photons m−2 s−1, Synechocystis cells recovered the dimension previously observed under the control photon irradiance (3.02 ± 0.13 μm). Altogether, cell size, growth rate and PSII functionality behaved similarly under increasing intensities of orange–red light, by reaching their maximal values at intensities up to 500 μmol photons m−2 s−1 and rapidly decreasing at photoinhibiting light conditions.

Table 1 Cell size and dry cell weight of Synechocystis grown under increasing light intensities.

The turbidostat-controlled cultures were grown at constant temperature (30° C) and pH (8.0) under orange-red light and acclimated for 24 h at each light intensity. The values are mean and standard deviation derived from three biological replicates. (*) Point of recovery at 200 µmol photons m−2 s−1.

Light intensity (μmol photons m−2 s−1)	Cell size (μm)	Dry cell weight (g OD730−1* L−1)	
50	2.11 ± 0.06	0.145 ± 0.001	
200	3.02 ± 0.13	0.140 ± 0.021	
300	3.22 ± 0.18	0.145 ± 0.005	
500	3.29 ± 0.13	0.147 ± 0.003	
800	3.09 ± 0.23	0.152 ± 0.004	
950	2.94 ± 0.26	0.153 ± 0.000	
1,460	2.69 ± 0.21	0.158 ± 0.021	
200*	3.03 ± 0.01	0.150 ± 0.003	

We then examined the effect of long-term acclimation to increasing intensities of orange–red light by estimating the Chl a content in Synechocystis through the measurement of the OD680/OD720 ratio (Table S2). Our measurements revealed the highest content of Chl a at 50 μmol photons m−2 s−1 compared to the other photon irradiances tested (Table S2). These data show that at limited light intensity Synechocystis sustains its growth by accumulating a high amount of Chl a to maximize light absorption. Increasing the photon irradiance to 200 μmol photons m−2 s−1, the amount of Chl a decreased by around 13%, which reached the minimum at 800 μmol photons m−2 s−1. Chl a reduction could be provoked by its synthesis inhibition in order to limit the absorption of harmfully excessive light (Kada et al., 2003; Xu et al., 2004). Unexpectedly, we observed Chl a increased also at 950 and 1,460 μmol photons m−2 s−1, which were shown to induce severe cell photoinhibition (Fig. 2).

Influence of local light intensity distribution on photoinibition of microorganism

Our multi-physics analysis focused on the spread of incident light within the Synechocystis cultivation apparatus and on the relationships between local light intensity and Synechocystis photosynthetic activity, which expectedly influence PBR productivity. The PBR used for culturing Synechocystis, the illumination setup and real-time monitoring are extensively described in Fig. 1, Table S3, Table S4 and Fig. S1. The in silico simulations have been developed under identical operational conditions as employed during the experiments, and are aimed at exploring the distribution of local light intensity within the cultivation apparatus as a function of the incident light intensity.

As thoroughly described in the Methods section, current knowledge on mass transfer, fluid dynamics, heat transfer from radiation, and growth kinetics were entirely incorporated to frame the modelling equations (Table S1), and the resulting model was implemented within the COMSOL 5.3® computing platform. To obtain suitable estimates for the parameters necessary to our mathematical representation, we relied either on experimentally determined values or on an inference procedure by fitting model simulations to observed data. The observed agreement between model estimates and experimental measurements for growth rate and dissolved oxygen (Figs. 2 and 3) demonstrated that our model is built on solid foundations. Moreover, we employed the light intensities recorded in our model simulations to estimate the light-dependent photosynthetic efficiency of the PBR in terms of moles of photons absorbed in the PBR per gram of biomass production, as shown in Table 2 and Table S5.

Table 2 Comparison of the efficiency of photosynthesis.

Comparison of the efficiency of photosynthesis in terms of moles of photons required for biomass production in Synechocystis calculated from growth simulations in a 380 ml vessel of the PBR upon acclimation for 24 h at 50, 200, 300, 500, 800 and 950 μmol photons m−2 s−1 of orange–red light. Shown are: Iin (μmol photons m−2 s−1), light intensity available to the PBR domains; μh−1, growth rate per hour; g DW L−1, biomass density in gram dry weight per liter; g DW/mol photons, growth yield in mol photons absorbed per gram biomass.

Iin (μmol photons m−2 s−1)	Vol L	μh−1	gDW L−1	mol photons g DW−1	Efficiency	
50	0.377	0.054	0.145	0.37	2.70	
200	0.377	0.110	0.140	0.61	1.64	
300	0.377	0.117	0.145	0.75	1.33	
500	0.377	0.110	0.147	1.11	0.90	
800	0.377	0.104	0.152	1.68	0.60	
950	0.377	0.088	0.153	2.10	0.48	

The observed agreement between model estimates and experimental measurements for growth rate and dissolved oxygen (Figs. 2 and 3) demonstrated that our model is built on solid foundations. Moreover, we employed the light intensities recorded in our model simulations to estimate the light-dependent photosynthetic efficiency of the PBR in terms of moles of photons absorbed in the PBR per gram of biomass production, as shown in Table 2.

These simulation results fully agree with experimental data acquired in previous studies (Schuurmans et al., 2015), and corroborated the plausibility of our modelling framework. The photosynthetic efficiency of Synechocystis turned out to decrease from the highest value, observed at 50 μmol photons m−2 s−1 where 2.70 g of biomass are produced per mol of photons, to the lowest value observed at 950 μmol photons m−2 s−1 where 0.48 g of biomass are produced per mol of photons. Moreover, the estimates of photosynthetic efficiency obtained by our model simulations confirmed that efficiency starts to drop fastest in the initial increase in intensity, which is expected at low OD batch cultures. Model simulations were then used to get insights into the local light intensity distribution along different combinations of directional axes within the cultivation apparatus. To ensure comparability of the results shown throughout our analysis, we normalized the light intensity values corresponding to the PBR internal space and resulting from our model simulations with respect to the initial incident intensity on the PBR surface, and we plotted the ratio between the calculated light intensity and the initial incident intensity RI=IcalcIinc

To analyse the distribution of light along all the directions, the trends of RI were developed by the YZ and XZ planes (Fig. 1D and Fig. S2). The light intensity distribution inside the liquid mixture on six YZ slices for three different incident light intensities, 50, 300 and 950 μmol photons m−2 s−1, shown in Fig. 4, confirms the ability of our modelling approach to capture a number of features of radiation spreading within a cultivation system. Boundary conditions and the coexistence of an upper gas phase and a lower liquid phase are expected to influence the light intensity distribution. Our modelling approach foresaw that the incident light gets reduced where the culture and glass of the vessel interface closest to the side where light is supplied. This reduction amounted to about 3–5%, and was more evident at lower intensities (Fig. 4). Moreover, in the area at the interface between the liquid and gas phases light intensity was found to display an eyelet-like pattern where the upper region shows higher intensity than the lower one. Additionally, this pattern was more remarkable for lower values of the incident light intensity. Another source of variation in the spatial distribution of light is identifiable in the rotating domain created by the stirring bar (Fig. S1). The local liquid movement propelled by the stirring bar rotation is expected to favour the light transmission process (Zhang, Dechatiwongse & Hellgardt, 2015). Our model simulations consistently predicted the increase of light intensity in the stirring bar neighbourhood, which amounted to around 3–5% for low Iinc and around 1–2% for high Iinc (Fig. 4C).

Figure 4 3D trend of normalized light intensity along YZ slices of the model PBR.

The calculated light intensity was normalized with respect to the initial incident intensity and the RI trend is performed along six YZ planes for three different incident light intensities, Iinc. (A) 50 μmol photons m−2 s−1. (B) 300 μmol photons m−2 s−1. (C) 950 μmol photons m−2 s−1.

In the interface area between the liquid phase and the bottom steel base, our model predicts a reduction in light intensity owing to the large difference between the absorption and emissivity values of these two domains. Beyond these effects, light intensity generally decreased with the distance from the light source, as shown by the plots at equally spaced slices along the YZ axis (Fig. 4) and further confirmed by the plots acquired at three XZ sections (Fig. 5 and Fig. S2).

Figure 5 2D trend of normalized light intensity along the XZ slices of the model PBR.

The calculated light intensity was normalized with respect to the initial incident intensity and the RI trend is performed along three XZ planes for three different incident light intensities, Iinc (A) 50 µmol photons m−2 s−1. (B) 300 µmol photons m−2 s−1. (C) 950 µmol photons m−2 s−1.

The light intensity decay along the path inside the cultivation apparatus was expected since it can be attributed to the compound effect of photon absorption by Synechocystis cells and of scattering phenomena (Grima et al., 1994). When varying the incident light intensity, our model simulations showed the extent of light decay to get lower at progressively higher incident light intensity. Decrease in light decay along the path amounting about 18% for Iinc of 50 μmol photons m−2 s−1, 12% for Iinc of 300 μmol photons m−2 s−1 and 7% for Iinc of 950 μmol photons m−2 s−1 were calculated. Our modelling framework was primed to afford the exploration of the relationship between the heterogeneity in local light distribution, which originates from the aforementioned sources, and the Synechocystis behaviour in the artificially lit cultivation system. More precisely, we expected the light environment created by the coexistence of regions of higher and lower local light intensity could favour Synechocystis functionalities, as gauged by the experimental measurements shown in Fig. 2.

From our model simulation results, it was evident that at 50 μmol photon m−2 s−1 the light penetration inside the culture was extremely low (Fig. 4A), suggesting that the cells were exposed to a period of lower intensity considerably longer than the expected light phase set experimentally. Under this light condition Synechocystis showed low growth rate, limited amount of oxygen dissolved in the medium (Fig. 2), and a higher Chl a amount than in the other tested conditions (Table S1), which is likely due to the need of optimizing light absorption in a scarcely lit environment. Our simulations showed that, upon increasing the irradiance to 300 μmol photon m−2 s−1, light managed to diffuse more deeply within the PBR interior (Figs. 4B and 5). The overall increased light availability and the coexistence of regions of different local light intensities created the most favourable light environment for Synechocystis which indeed reached its maximal growth rate and oxygen dissolved in the medium (Fig. 2). Conversely, the further rise of light intensity up to 950 μmol photons m−2 s−1, and the subsequent suppression of low local light intensity experienced by the cells (Figs. 4C and 5), led Synechocystis to suffer high light stress reflected in a noticeable decrease in its growth rate and oxygen evolution activity (Fig. 2). Under such light conditions, cells were no longer able to dissipate the excess of supplied light and entered a photoinhibition state (Chiang, Lee & Chen, 2011).

Discussion

The experiments conducted were particularly informative for studying the effects of increasing intensities, ranging between 50 and 1,460 μmol photons m−2 s−1, of red–orange light on the adjustments of the physiological state in Synechocystis. Although the PSII photodamage effect caused by the exposure to high (white) light intensity is well-established in plants (Theis & Schroda, 2016), corresponding quantitative data on photoinhibition by red light in cyanobacteria are scarce. Furthermore, experimental evidence of the recovery potential in Synechocystis from this light stress has so far been lacking. Importantly, our results evidenced the adaptive capacity of Synechocystis to completely recover from the harmful condition of photoinhibition under subsequent exposure to an optimal light intensity for growth. It is already well known that under photoinhibitory conditions, loss of oxygen evolution capacity of PSII activates the PSII repair cycle and that the rate of the repair reaction depends on the extent of the PSII damaged centres (Tyystjärvi, 2013). Since the D1 protein is the main target of photoinhibition and its lack within the PSII centre speeds up its synthesis, an increase of the amount of psbA transcript in Synechocystis occurs in high light (Mohamed et al., 1993) and this high level is maintained for several hours in darkness (He & Vermaas, 1998). Here the quick recovery of PSII activity observed at 200 μmol photons m−2 s−1 suggests that the high levels of D1 transcript maintained within the surviving cells at 1,460 μmol photons m−2 s−1 allows the acceleration of D1 synthesis to reactivate the PSII activity, once cells have been reverted to the optimal growth irradiance. An exceptional capacity to cope with fluctuations in a wide range of lights differing for spectral quality and quantity, as well as pH and temperature, was previously observed in this microorganism by Constant et al. (2000) and Zavřel et al. (2015). Indeed, the ability to readily adapt the metabolism to different environmental conditions allowed Synechocystis, and more in general cyanobacteria, to proliferate even in extreme environments on Earth (Hernández-Prieto et al., 2016).

Furthermore, growth rate was found to positively correlate with cell size independently of light intensity (Fig. 6). This trend has been previously observed in Synechocystis acclimated to lower intensities of light (Du et al., 2016) and in Synechococcus grown under limited nutrient supply of phosphate and nitrate (Garcia, Bonachela & Martiny, 2016), and hints at a tight coordination of growth rate with cell size. It has been reported that cyanobacteria can respond to different abiotic stresses by increasing the cell size, accumulating granules of different nature, e.g. glycogen, upon exposure to high light intensity (Kopecna et al., 2012) and polyhydroxybutyrate in case of high temperatures (Červený et al., 2015). Even though the mechanisms that associate cellular growth rate with cell size are still unclear (Amir, 2014), here we suggest that the increase of light intensity could accelerate both the metabolism and growth rate of Synechocystis, favouring the accumulation of higher amounts of biomass that need to be properly stored, leading the cells to increase their size (Amir, 2014; Ferrezuelo et al., 2012). In general, the variation in cell size observed at incremental steps of light intensity was accompanied by the maintenance of a relatively constant dry cell weight per OD730 (Table 1), which suggests that the fluctuations observed in cell size exposed to increasing light intensities were counterbalanced by opposite fluctuations in the number of cells per volume.

Figure 6 Relationship of the cell size and growth rate of Synechocystis set at different light intensities.

The figure displays the relationship between growth rate and cell size at varying light intensities. Dot colours reflect the incident light intensity, with the green and red colours corresponding to low values and high values, respectively.

The highest content of Chl a at 50 μmol photons m−2 s−1 along with its decrease at increasing orange–red light intensities which we observed upon increasing the orange–red light intensity are expected since inhibition of Chl a synthesis could serve to limit the absorption of harmfully excessive light. Conversely, the Chl a increase observed at 950 and 1,460 μmol photons m−2 s−1, which were shown to induce severe cell photoinhibition (Fig. 2), is unexpected. This observation warrants further investigation but it is plausible to hypothesize that the observed increase in Chl a amount could be partially due to the accumulation of this pigment in dead cells, which likely reflects the much longer lifetime of Chl molecules with respect to that of other pigments (Steiger, Schäfer & Sandmann, 1999; Vavilin, Brune & Vermaas, 2005; Vavilin & Vermaas, 2007; Yao et al., 2012; Yao, Brune & Vermaas, 2012; Trautmann, Beyer & Al-Babili, 2013; Havaux, 2014). Finally, the recovery of pigment biosynthesis observed upon acclimation of photoinhibited cells over 24 h at 200 μmol photons m−2 s−1 confirmed the elevated degree of plasticity of this cyanobacterium to cope with extremely high light intensities.

The in silico simulations were focused to investigate the spatial distribution of local light intensity in the cultivation system on the basis of the incident light intensity, and our model simulation allowed a careful, albeit qualitative, evaluation of the complex consequences that variations in light intensity and local light distribution can cause on the Synechocystis physiology within an artificially lit cultivation system. In particular, we were able to identify three distinct operative states: (i) a light limited state where all the light supplied to the system is maximally exploited by the cells, a condition that is reflected in a linear relation between irradiance and Synechocystis physiological parameters, (ii) a light optimal state where Synechocystis optimizes the utilization of light to support its maximal growth rate and photosynthetic activity, (iii) a photo-inhibition state where the excess of incident light becomes harmful for microorganisms growth. Our model simulations suggest that regulating the incident light on the PBR, at least in a range of moderate intensities, could be used to enhance Synechocystis growth. Indeed, this model takes into account the formation of areas of different local light intensity within the PBR, whose extent varies as a function of the incident light intensity, and that can be exploited by the microorganism to prevent from experiencing light-induced stress. According to our study, managing local light effects is expected to be worth careful consideration in PBR design for leveraging the microorganism exploitation.

Conclusions

The productivity of PBRs exploiting Synechocystis clearly depends on the photosynthetic efficiency of this microorganism. Since this efficiency largely depends on the cyanobacteria ability to manage the light collected in the cultivation apparatus, in this work we thoroughly investigated the impact of the setup of the light conditions in the PBR on Synechocystis growth and photosynthetic activity. Monitoring Synechocystis’ physiological state under increasing intensities of orange–red light, we found that growth rate, cell size and PSII activity were influenced by light intensity, although in slightly different ways. Synechocystis cells proved to be resistant to high light stress conditions, showing photoinhibition only above 800 μmol photons m−2 s−1 combined with a remarkable ability to recover from the complete state of photoinhibition experienced at 1,460 μmol photons m−2 s−1 when reverting light to 200 μmol photons m−2 s−1. Considering the notable plasticity of Synechocystis in response to changes in light intensity, we searched for unknown features of the PBR light conditions that could leverage Synechocystis behavioural features to enhance the overall PBR productivity. To this end, we deemed it particularly useful to adopt also an in silico methodology by constructing a PBR model and subsequently use it to simulate the effects of increasing incident light intensities on the local light intensity distribution. Interestingly, our results indicate that the formation of areas of different light intensities could be controllable by tuning the incident light intensity on the PBR. A gain in Synechocystis viability is achievable by increasing the incident light intensity as far as areas of different local light intensities exist to allow Synechocystis cells to escape from the photoinhibition state. It is useful to note that the observations herein presented are drawn from experiments and simulations carried out in turbidostat mode (constant OD730) and could vary depending on the choice of the cultivation mode. Nonetheless, our results provide useful insights in a PBR modelling perspective and, in particular, suggest that a PBR design would benefit from considering the management of local light heterogeneity to increase the microorganism photosynthetic activity, by limiting photoinhibition phenomena, to ultimately maximize the productivity.

Supplemental Information

Supplemental Information 1 Supplementary Information including supplementary figures and tables.

This file reports additional figures displaying slice trend of dispersed phase, photobioreactor characteristics, full set of equations employed in the modelling approach and additional physiological features for Synechocystis.

Click here for additional data file.

Supplemental Information 2 Raw data reporting photobioreactor setting variables and numerical data enabling the quantification of Synechocystis physiological features.

This file reports photobioreactor setting variables and numerical data enabling quantification of Synechocystis physiological features. This file refers to the biological replicate no. 2.

Click here for additional data file.

Supplemental Information 3 Raw data reporting photobioreactor setting variables and numerical data enabling quantification of Synechocystis physiological features.

This file reports the photobioreactor setting variables and numerical data enabling quantification of Synechocystis physiological features. This fiule refers to the biological replicate no. 1.

Click here for additional data file.

Supplemental Information 4 Raw data reporting photobioreactor setting variables and numerical data enabling quantification of Synechocystis physiological features.

This file reports photobioreactor setting variables and numerical data enabling quantification of Synechocystis physiological features. This referes to the biological replicate no. 3.

Click here for additional data file.

Nomenclature

A area, m2

cd mass fraction of dispersed phase, kg kg−1

C concentration, mol m−3

Cp specific heat at constant pressure, J m−3 K−1

D diffusion coefficients, m2 s−1

Dmd turbulent dispersion coefficient, m2 s−1

e enthalpy flux density, J m−2 s−1

EA activation energy, J mol−1

F force term, kg m−2 s−2

G incident light radiation, W m−2

hj(T) enthalpies heat flux densities, J m−2 s−1

I incident light intensity, W m−2

Ib black body radiation, Wm−2

J diffusion vector

k turbulent kinetic energy, m2 s−3

Kr reaction rate constant, m2 s−1

m mass of species, kg

mdc mass transfer from dispersed to continuous phase, kg m−3s−1

M molar mass, kg mol−1

n flux density, mol m−2 s−1

nd relative mass flux, mol m−2 s−1

p pressure, Pa

q heat flux densities, W m−2

Q volumetric charge density, C m−3

Qr radiative flux, W m−2

R universal gas constant, J K−1 mol−1

T temperature, K

u velocity vector, m s−1

uc continuous phase velocity vector, m s−1

ud dispersed phase velocity vector, m s−1

uslip slip velocity vector, m s−1

νp the convective velocity, m s−1

w volume fraction

x mass fraction

Greek symbols

β extinction coefficient, m−1

ε turbulent energy dissipation, m2s−3

kc effective thermal conductivity coefficient, W m−1 K−1

κ absorbance coefficient, m−1

μ dynamic viscosity, kgf s m−2

μgr growth rate, h−1

μT turbulent viscosity,

ν stoichiometric coefficients

ρ density, Kg m−3

σS scattering coefficient, m−1

τ turbulent stress, -

ϕc continuous phase fraction, -

ϕd dispersed phase fraction, -

ω rotational velocity, rad s−1

Additional Information and Declarations

Competing Interests

Author Contributions

Data Availability

The authors declare that they have no competing interests.

Alessandro Cordara performed the experiments, prepared figures and/or tables, approved the final draft.

Angela Re analysed the data, authored or reviewed drafts of the paper, approved the final draft.

Cristina Pagliano analysed the data, prepared figures and/or tables, approved the final draft.

Pascal Van Alphen conceived and designed the experiments, analysed the data, contributed reagents/materials/analysis tools, approved the final draft.

Raffaele Pirone authored or reviewed drafts of the paper, approved the final draft.

Guido Saracco contributed reagents/materials/analysis tools, approved the final draft.

Filipe Branco dos Santos authored or reviewed drafts of the paper, approved the final draft.

Klaas Hellingwerf conceived and designed the experiments, analysed the data, contributed reagents/materials/analysis tools, approved the final draft.

Nicolò Vasile performed the experiments, prepared figures and/or tables, approved the final draft.

The following information was supplied regarding data availability:

The raw data are provided in the Supplemental Files.

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
