# Peer review of "Analysis of the light intensity dependence of the growth of Synechocystis and of the light distribution in a photobioreactor energized by 635 nm light"

_PeerJ, doi:10.7717/peerj.5256_

## Round 0.1 · original submission · Major Revisions

Thank you for your submission to PeerJ. Your manuscript is well written and very nicely presented with high technical standards. However, the reviewers have indicated that it needs several major and minor improvements in line with the recommendations given within the listed referee reports. I also would like to advise you to enrich the discussions for highlighting the novelty of your work as mainly pointed out by the 3rd referee. Please resubmit your revised manuscript after carefully addressing the issues raised below. Once again, thank you for submitting your manuscript to PeerJ and I look forward to receiving your revision.

Reviewer 1 ·

Basic reporting

no comment

Experimental design

no comment

Validity of the findings

no comment

Additional comments

The growth rate, cell size and photosynthetic activity were investigated at various orange-red light intensities. The local light intensity and light regime in the PBR were further simulated at different incident light intensity. The work is interesting, and some useful and valuable results were obtained.
Only a little shortage as follows:
In Line 469, as well as other place, the unit of cell size is wrong, it should be μm rather than m.
In Figure 6, what is the detail position of three XZ planes? It should be showed.

Reviewer 2 ·

Basic reporting

In this work, authors describe the investigation of the effect of increasing intensities of orange-red light on the growth of Synechocystis in a laboratory scale photobioreator. The article is clear and well organized.
Regarding references:
Lines: 723, 754, 780, 803, 822, 882 and 958 please add the missing information to these articles. In some cases the Journal title is missing in other the year and the volume/page numbers.
Line 809: article title is repeated for three times, please correct.
Please italicize species name.
Figures 2 and 4. Please replace commas with points.
Table S4. Please correct the gravity force value. Please add unit for incident light intensity.

Experimental design

No comment

Validity of the findings

No comment

Additional comments

The "μ", for micro, letter disappeared in several points in the manuscript. I presume that the symbol is lost with the conversion from .doc to .pdf file. Please check and correct these points.

Reviewer 3 ·

Basic reporting

See general comments

Experimental design

See general comments

Validity of the findings

See general comments

Additional comments

Dear editor, dear authors,

I read the manuscript ‘Managing light distribution effects for leveraging photosynthetic activity in photobioreactors’ with great interest. The manuscript is clearly written and easily readable. Figures are of very high quality and advanced modelling tools are used. The experimental dataset generated is valuable as it seems to be applicable for advanced modelling studies. Nevertheless I am not convinced about the novelty of this study because I did not find any results which yield new insights in photoautotrophic microbial growth. For this reason I recommend not to accept this manuscript for publication in PeerJ. Below I will go in more detail on how I came to this conclusion.

The title of the manuscript for me was a bit misleading because in the end there was nothing about ‘managing’ light distribution. The issue of managing light distribution clearly is an important topic for large-scale application of such bioprocesses. I think that this experimental dataset and developed modelling tools can be used in a completely new study really focussed on this topic.

Now I will go further on my conclusion of lack of novelty. The main findings of the experimental work is the finding that growth of Synechocystis is light limited at low light intensities, saturates reaching maximal specific growth rate at increasing light intensities, and decreases again at very high light levels. This is a well-known response for all photoautotrophic microbes (algae and cyanobacteria) grown under nutrient replete conditions and this is not novel. Only the observation that Synechocystis could quickly recover from photoinhibition could be new. But in this context I miss an in-depth discussion on recovery from photoinhibition, for example the relation to the photosystem II damage-repair cycle (see reference below). I think a literature review on this topic probably will show that a rapid recovery to photoinhibition is a much more general feature of microalgae and cyanobacteria.
Theis, J. and Schroda, M. (2016) ‘Revisiting the photosystem II repair cycle’, Plant Signaling and Behavior. Taylor & Francis, 11(9), pp. 1–8. doi: 10.1080/15592324.2016.1218587.

In this manuscript modelling of fluid and gas dynamics and light regime in the photobioreactor is very extensively described. But, also in this part of the study I did not recognize the novelty. In fact, it is not clear to me why modelling fluid and gas flow, and heat transfer was relevant anyway. In the end the only conclusion coming from the modelling study was the observation that light distribution in the culture was heterogeneous. This is widely recognized. The heterogeneity in the y-direction (figure 5) is implicitly present in the Lambert Beer equation. So, there is no need for model calculations because a qualitative evaluation of equation 16 is sufficient to arrive at this conclusion. The heterogeneity of light over the light exposed reactor surface is a direct result of the design of the light source and photobioreactor, and also here complex modelling is not really needed; one could just measure the light distribution over the surface. The lack of use of the models (especially fluid and gas dynamics) contrasts greatly with the extensive description of these models in the manuscript.

At this point I would like to stress that modelling of the liquid flow is relevant in case one wants to apply a dynamic photosynthesis model. For example a model which takes into account photoinhibition in zones with strong illumination and recovery from photoinhibition in dark zones. Please refer to below publication in which such an approach was used. But, such an approach was not taken in the current study so I do not see the use of CFD modelling.
Olivieri, G., Gargiulo, L., Lettieri, P., Mazzei, L., Salatino, P. and Marzocchella, A. (2015) ‘Photobioreactors for microalgal cultures: A Lagrangian model coupling hydrodynamics and kinetics’, Biotechnology Progress, 31(5), pp. 1259–1272. doi: 10.1002/btpr.2138.

In the calculation of the biomass yield on photons I suspect a mistake. A yield of 2.70 g/mol was reported (Table 2). Such a high yield is extremely unlikely as it would mean a minimal quantum requirement of 8 photons per O2 and (!) the production of merely carbohydrates instead of functional biomass, which is not very likely at such low light levels. To my knowledge yields higher than 2 g/mol have not been reported up to date (see for example reference to Schuurmans below). Re-examining the data presented in table 2 allowed me to recalculate the biomass yield on light. Assuming that the optical depth of the PBR is 2.5 cm and assuming that all light is absorbed in the culture I calculated a volumetric light input of 7.2 mol/m3/h (at 50 micromol/m2/s). The volumetric biomass productivity was 8.1 g/m3/h (at 50 micromol/m2/s). Dividing 8.1 by 7.2 gives a biomass yield of 1.125 g/mol, which corresponds to published data. In this context it is relevant to know the light intensity at the backside of the reactor. Was this measured? (I could not find information in supplementary data).
Schuurmans, R. M., Van Alphen, P., Schuurmans, J. M., Matthijs, H. C. P. and Hellingwerf, K. J. (2015) ‘Comparison of the photosynthetic yield of cyanobacteria and green algae: Different methods give different answers’, PLoS ONE, 10(9), pp. 1–17. doi: 10.1371/journal.pone.0139061.

Minor comments:

In my opinion the discussion of the relation between cell size and specific growth rate should be extended with the possible influence of a persistent circadian rhythm (see reference below). Could such a rhythm have had an impact on the measurements? Was there an change of average cell size over a 24 hours
van Alphen, P. and Hellingwerf, K. J. (2015) ‘Sustained Circadian Rhythms in Continuous Light in Synechocystis sp. PCC6803 Growing in a Well-Controlled Photobioreactor’, Plos One, 10(6), p. e0127715. doi: 10.1371/journal.pone.0127715.period?

With respect to the materials and methods the procedure to measure the specific growth rate could be explained in more detail. As I understand every 24 hours light levels were increased. So within 24 hours the cyanobacteria have to acclimate and reach a new steady state, and then the specific growth rate is determined. Is this possible in only 24 hours? Is it really possible to state that a steady state was reached? If not, what is the possible impact on the data obtained?

Lines 158-159 are not clear to me. How was the turbidostat condition maintained when increasing light levels?

Lines 153-154 are conflicting: pCO2 was adjusted and at the same time CO2 was always at 1% v/v.

Model description is not clear to me with respect to the gas phase, specifically lines 228-229 and lines 241-242. I think this aspect has to be described more extensively.

Assessment of different model parameters is not clearly described. For example gas bubble size in equation 18.

Was the general radiative transfer equation really used? If so, for what purpose? Light regime was modelled with equation 18 I think?

What has been done with heat transfer (equation 20-21)? Is it relevant to model heat transfer since temperature is maintained constant by the FMT150 reactor unit.

---

## Round 0.2 · accepted · Accept

Thank you for your revised re-submission package. I am pleased to inform you that your manuscript, entitled as "Analysis of the light intensity dependence of the growth of Synechocystis and of the light distribution in a photobioreactor energized by 635 nm light” has been accepted for publication in PeerJ. Thank you for submitting your work to PeerJ. We look forward to seeing more of your work in the future.

#